# Heterogeneity coordinates bacterial multi-gene expression in single cells

**Yichao Han**[1], **Fuzhong Zhang**[1,2,3] *

**1** Department of Energy, Environmental and Chemical Engineering, Washington University in St. Louis, Saint Louis, Missouri, United States of America, **2** Division of Biological & Biomedical Sciences, Washington University in St. Louis, Saint Louis, Missouri, United States of America, **3** Institute of Materials Science & Engineering, Washington University in St. Louis, Saint Louis, Missouri, United States of America

* fzhang@seas.wustl.edu

**Data Availability Statement:** All relevant data are within the manuscript and its Supporting Information files.

**Funding:** This work was supported by National Science Foundation (MCB1453147 to FZ), Human Frontier Science Program (RGY0076/2015 to FZ),

## Abstract

For a genetically identical microbial population, multi-gene expression in various environments requires effective allocation of limited resources and precise control of heterogeneity among individual cells. However, it is unclear how resource allocation and cell-to-cell variation jointly shape the overall performance. Here we demonstrate a Simpson's paradox during overexpression of multiple genes: two competing proteins in single cells correlated positively for every induction condition, but the overall correlation was negative. Yet this phenomenon was not observed between two competing mRNAs in single cells. Our analytical framework shows that the phenomenon arises from competition for translational resource, with the correlation modulated by both mRNA and ribosome variability. Thus, heterogeneity plays a key role in single-cell multi-gene expression and provides the population with an evolutionary advantage, as demonstrated in this study.

## Author summary

Microbes perform multitasking for a wide range of purposes, including survival, adaptation, colonization, and evolution. Both modelling and experimental results at the ensemble level reveal trade-offs between different tasks due to resource competition, but it is unclear how single cells allocate limited intracellular resources to perform multitasking, and how does a population coordinate single cell performances during multitasking to maximize population efficiencies. In this study, we address this question by using bacterial multi-gene overexpression as the basic form of multitasking. We discovered and analyzed a statistical phenomenon called Simpson's paradox, where competing proteins in single cells correlate positively at each constant condition, although the proteins correlate negatively when all conditions are combined. We demonstrate that the phenomenon arises from competition for translational resources, with the correlation modulated by heterogeneity of both mRNA and ribosomes. We further show that heterogeneity coordinates multiple functional modules, conferring an evolutionary advantage on the population. Our work discloses that heterogeneity in the form of Simpson's paradox is an important phenomenon in coordinating multi-gene expression.

and National Institute of General Medical Sciences of the National Institutes of Health (R35GM133797 to FZ). YH is supported by the T32 HG000045 training grant from the National Human Genome Research Institute. The content is solely the responsibility of the authors and does not necessarily represent the official views of the funding agencies. The funders had no role in study design, data collection and analysis, decision to publish, or preparation of the manuscript.

**Competing interests:** The authors have declared that no competing interests exist.

## Introduction

Bacteria often simultaneously turn on the expression of multiple pathways or cellular machineries to perform multitasking in response to various conditions. Obtaining optimal outcomes of multitasking is critical for population survival, bacteria-host interaction, cell-to-cell communication, biofilm formation, and biosynthetic performance [1–5]. During multitasking, modules for different tasks often compete with each other for limited intracellular resources, which could affect the performance of the overall system [6–9]. At the most fundamental level, it has been widely observed that overexpression of a heterologous gene decreases the expression level of other genes, leading to a negative correlation between competing proteins at the ensemble level [10–12]. Meanwhile, the performance of a module also varies from cell to cell due to biological stochasticity, leading to phenotypic heterogeneity. Distinctive phenotypes within a genetically identical population are sometimes harnessed as a mechanism for division of labor, where distinct subpopulations perform different tasks, thus reducing resource competition within each single cell. However, it remains elusive to what degree phenotypic heterogeneity affects simultaneous operation of multiple functional modules within every single cell. Specifically, how do single cells deal with resource competition, and how does a population coordinate single cell performances during multitasking to maximize population efficiencies [2,13,14]?

## Results

In bacteria, RNA polymerases (RNAPs) and ribosomes are believed to be the limiting factors of transcription and translation, respectively [15]. To examine single cell multitasking in the most fundamental form, we designed two competing gene overexpression modules with fluorescent proteins as outputs (Fig 1A). One of them contains a constitutively expressed green fluorescent protein (*gfp*) gene in the *Escherichia coli* chromosome mimicking a naturally-occurring module [11]. The other competing module contains a *Mycobacterium marinum* carboxylic acid reductase (*car*) gene fused with an *mCherry* gene in a medium-copy plasmid. In our test *E. coli* strain, the burdensome CAR-mCherry protein does not serve any additional cellular or metabolic function [16], except for consuming global resources for both transcription and translation during its expression. Isopropyl β-D-1-thiogalactopyranoside (IPTG) mimics an environmental signal to increase the output of this module. Single cell GFP and CAR-mCherry fluorescence in steady state conditions was measured using fluorescence microscopy (Fig 1B) to evaluate heterogeneity in cellular performance. Under different IPTG conditions, the population mean GFP fluorescence decreased as the population mean CAR-mCherry fluorescence increased (Fig 1C), suggesting the presence of resource competition between the two proteins, in good agreement with previous ensemble-level observations [11,12]. At the single-cell level, the joint distribution of GFP and CAR-mCherry proteins resembled a statistical phenomenon called Simpson's paradox [17]: the correlations between GFP and CAR-mCherry in single cells were positive at each constant induction condition, whereas the overall correlation became negative when the data for all induction conditions were merged (Fig 1D and S1A Fig). The negative trend is not affected by sample sizes when merged data is evenly sampled across induction conditions, and the standard deviation of correlation decreases with larger sample size (S1B Fig). The merged condition exemplifies the heterogenous and fluctuating environments where a microbial community lives, while each induction condition exemplifies constant environments that a local microbial group adapts. Thus, Simpson's paradox phenomenon in bacterial gene expression may present in multiple systems where local regions have relative consistent module inputs while these inputs vary

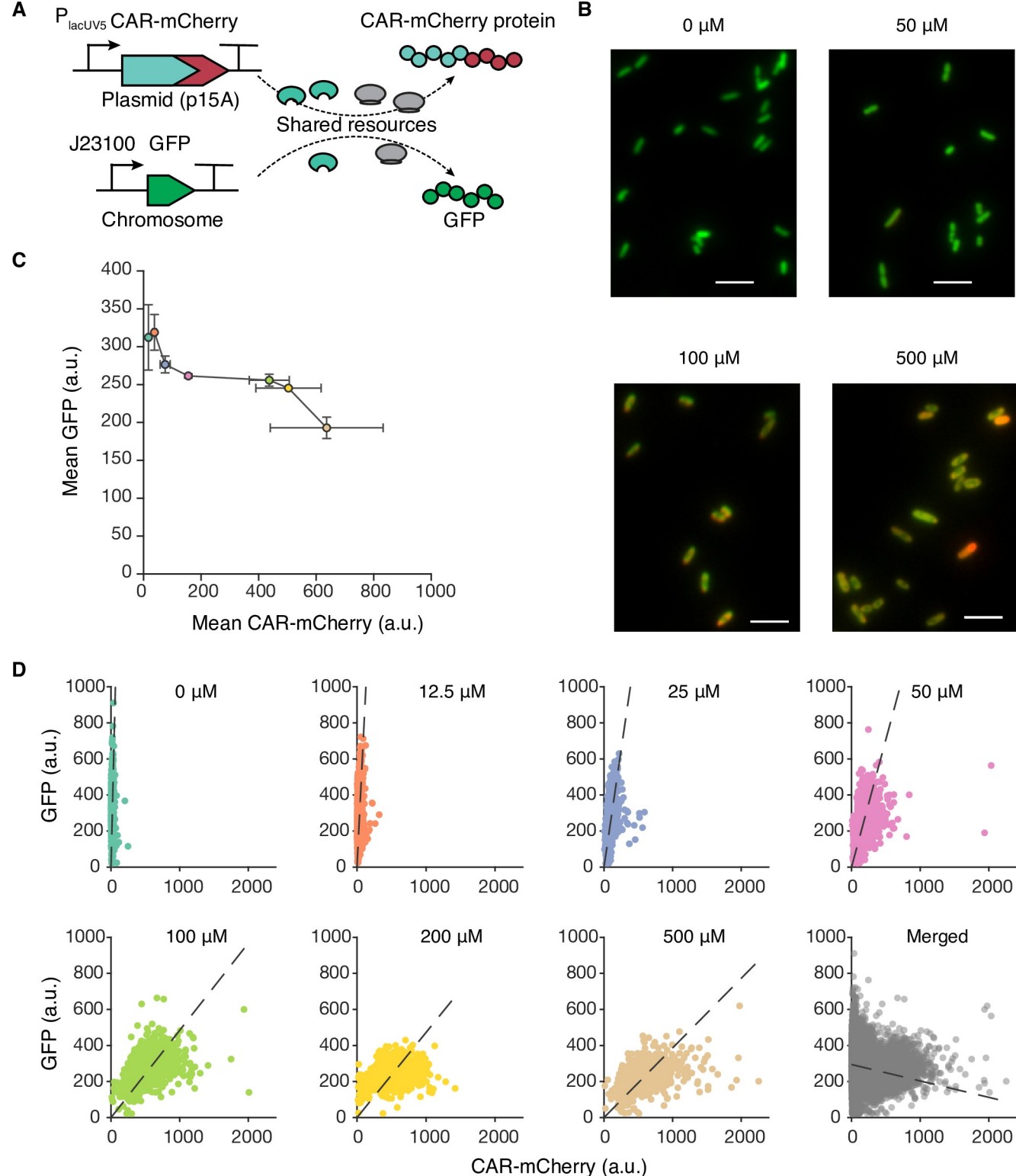

**Fig 1. Multi-gene expression in single-cells during translational competition.** (**A**) Translational competition of CAR-mCherry and GFP over limited shared ribosomes in single cells. The CAR-mCherry mRNAs are transcribed from an IPTG-inducible $P_{lacUV5}$ promoter, while GFP mRNAs are constitutively transcribed. (**B**) Representative fluorescence images of combined green (GFP) and red (CAR-mCherry) channels at various induction levels. IPTG concentrations are labelled at the top of each image. Scale bars, 5 μm. (**C**) Population mean fluorescent intensity of GFP and CAR-mCherry at various IPTG induction levels. Error bars represent standard deviations of three replicates from different days. (**D**) Correlation between CAR-mCherry and GFP

expression levels of single cells at various IPTG induction levels. The last plot contains all data points merged from the other seven plots. The dashed lines represent linear fittings to the data. a.u., arbitrary units.

significantly among different regions in the system, such as biofilms [18] or large-scale fermenters [14]. The opposite correlation patterns suggest that a microbial community has the potential to explore a large area of protein expression space within the resource-limiting region and balance the outcome of multiple tasks (e.g., a certain ratio of correlated protein expression) according to the local environment.

To understand the observed Simpson's paradox and to quantify the combined effects of both resource competition and cell-to-cell variation on multi-gene overexpression, we developed a generic analytic framework that can be applied to resource competition at different levels (e.g., transcription, translation, and metabolism). Compared to previous resource competition models [7,11,19–22], our model considers cell-to-cell variations in resource availability and focuses on heterologous expression systems that have strong competition with the endogenous expression system, thus uniquely illuminating resource competition in engineered cells at the single-cell level [23,24]. Our model has several important assumptions: i) to emphasize the effect of resource competition, the two competing modules do not shared transcriptional nor translational regulators, such as transcription factors and small RNAs; ii) the amounts of resource available for gene expression, such as RNA polymerase or ribosome, vary among single cells; and iii) all macroscopic reaction rate constants are evaluated at steady state and do not vary among single cells.

The model was first applied to study translational competition (Note 1 in S1 Text), where two module inputs, total heterologous mRNAs ($M_1^T$) and total endogenous mRNAs ($M_2^T$), compete for the limited amount of total ribosomes ($Rib^T$), and produce heterologous proteins ($P_1$) and endogenous proteins ($P_2$), respectively (Fig 2A). When $Rib^T$ inside an individual cell is fixed,

$$Rib^T = Rib^F + \frac{n_1\ Rib^F}{\beta_1 + Rib^F}M_1^T + \frac{n_2\ Rib^F}{\beta_2 + Rib^F}M_2^T,\tag{1}$$

where $Rib^F$ is the number of free ribosomes, $n_i$ is the average number of ribosomes bound to the corresponding mRNA (i = 1, 2), and $\beta_i$ represents the dissociation constant. On the right side, the second term $\frac{n_1\ Rib^F}{\beta_1 + Rib^F}M_1^T$ is proportional to $P_1$, and the third term $\frac{n_2\ Rib^F}{\beta_2 + Rib^F}M_2^T$ is proportional to $P_2$. The repression on $P_2$ caused by increasing $M_1^T$ ($\frac{\partial P_2}{\partial M_1^T}$) indicates the strength of resource competition. In each cell, lower $Rib^T$ and higher $M_1^T$ create stronger competition due to fewer $Rib^F$ (Fig 2B). The dissociation constants $\beta_1$ and $\beta_2$ largely determine $\frac{\partial Rib^F}{\partial M_1^T}$ and $\frac{\partial P_2}{\partial Rib^F}$ respectively (Note1 in S1 Text). If $\beta_1$ is much larger than $Rib^F$, the heterologous proteins $P_1$ are not burdensome enough to sequester a significant amount of free ribosomes (i.e. the absolute value of $\frac{\partial Rib^F}{\partial M_1^T}$ is small). If $\beta_2$ is much smaller than $Rib^F$, the expression of endogenous proteins $P_2$ are not affected by reduced $Rib^F$ (i.e. the value of $\frac{\partial P_2}{\partial Rib^F}$ is small). In both cases, the strength of resource competition is negligible (S2A and S2B Fig).

To introduce cell-to-cell variations, $M_1^T$, $M_2^T$, and $Rib^T$ are considered as random variables for individual cells, although they are assumed to be constants over time for each cell. At steady

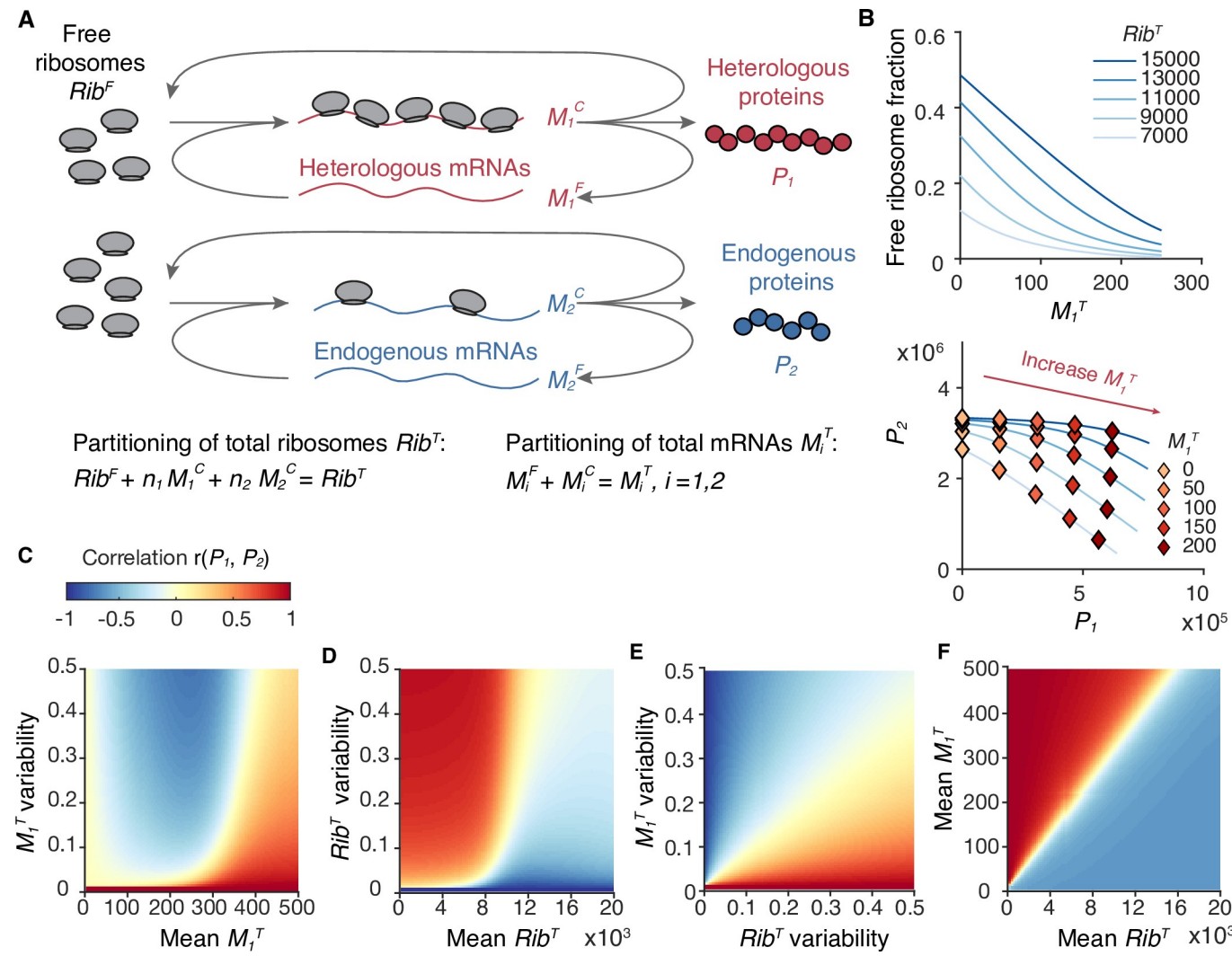

**Fig 2. Coarse-grained model of translational resource competition. (A)** The coarse-grained model considers ribosome allocation between heterologous ($i = 1$) and endogenous ($i = 2$) mRNAs. The input, the output, and the resource are total mRNA $M_i^T$, protein $P_i$, and total ribosome $Rib^T$, respectively. $Rib^T$ can either be free ribosome $Rib^F$ or mRNA-bound ribosome. **(B)** Ribosome competition in a single cell. Top, decrement of the free ribosome fraction ($Rib^F/Rib^T$) caused by increasing $M_1^T$. Bottom, negative correlation between endogenous protein ($P_2$) and heterologous proteins ($P_1$). Calculations of $Rib^F$, $P_1$, and $P_2$ are described in Note 1.2 in S1 Text, with parameters listed in Table A in S1 Text. **(C-F)** Correlation between $P_1$ and $P_2$ of single cells, r($P_1$, $P_2$). Calculation of r($P_1$, $P_2$) is described in Note 1.3 in S1 Text. $M_2^T$ variability is set as zero for simplicity. **(C)** Mean $Rib^T$ (10,000) and $Rib^T$ variability (0.1) are set as constants. **(D)** Mean $M_1^T$ (300) and $M_1^T$ variability (0.1) are set as constants. **(E)** Mean $M_1^T$ (300) and mean $Rib^T$ (10,000) are set as constants. **(F)** $M_1^T$ variability and $Rib^T$ variability (both 0.1) are set as constants.

state, cell-to-cell variations of protein expression levels can be described by a linearized model:

$$
\begin{pmatrix} P_1 \\ P_2 \end{pmatrix} = \begin{pmatrix} \overline{P_1} \\ \overline{P_2} \end{pmatrix} + \begin{pmatrix} \dfrac{\partial P_1}{\partial M_1^T} & \dfrac{\partial P_1}{\partial M_2^T} & \dfrac{\partial P_1}{\partial Rib^T} \\ \dfrac{\partial P_2}{\partial M_1^T} & \dfrac{\partial P_2}{\partial M_2^T} & \dfrac{\partial P_2}{\partial Rib^T} \end{pmatrix} \begin{pmatrix} M_1^T - \overline{M_1^T} \\ M_2^T - \overline{M_2^T} \\ Rib^T - \overline{Rib^T} \end{pmatrix},
$$

(2)

where $\overline{X}$ denotes the mean value of $X$ at steady state. The covariance between $P_1$ and $P_2$ at

steady state is derived as

$$Cov(P_1, P_2) = \frac{\partial P_1}{\partial M_1^T}\frac{\partial P_2}{\partial M_1^T}Var(M_1^T) + \frac{\partial P_1}{\partial M_2^T}\frac{\partial P_2}{\partial M_2^T}Var(M_2^T) + \frac{\partial P_1}{\partial Rib^T}\frac{\partial P_2}{\partial Rib^T}Var(Rib^T)$$

$$+ \left(\frac{\partial P_1}{\partial M_1^T}\frac{\partial P_2}{\partial M_2^T} + \frac{\partial P_2}{\partial M_1^T}\frac{\partial P_1}{\partial M_2^T}\right)Cov(M_1^T, M_2^T)$$

$$+ \left(\frac{\partial P_1}{\partial M_1^T}\frac{\partial P_2}{\partial Rib^T} + \frac{\partial P_2}{\partial M_1^T}\frac{\partial P_1}{\partial Rib^T}\right)Cov(M_1^T, Rib^T)$$

$$+ \left(\frac{\partial P_1}{\partial M_2^T}\frac{\partial P_2}{\partial Rib^T} + \frac{\partial P_2}{\partial M_2^T}\frac{\partial P_1}{\partial Rib^T}\right)Cov(M_2^T, Rib^T), \tag{3}$$

Considering the cell-to-cell variations in $Rib^T$ and $M_1^T$ as the two main sources of cellular heterogeneity in this system, the covariance between $P_1$ and $P_2$ at steady state can be further approximated as a linear combination of the variances in $Rib^T$ and $M_1^T$:

$$\text{Cov}(P_1, P_2) = \frac{\partial P_1}{\partial Rib^T}\frac{\partial P_2}{\partial Rib^T}\text{Var}(Rib^T) + \frac{\partial P_1}{\partial M_1^T}\frac{\partial P_2}{\partial M_1^T}\text{Var}(M_1^T), \tag{4}$$

where the first term is positive, and the second term is negative due to the competition effect ($\frac{\partial P_2}{\partial M_1^T} < 0$). Critically, the opposite contributions from variances in $Rib^T$ and $M_1^T$ reveal that variation in the shared resource strengthens the correlation of module outputs, whereas variation in the competing module inputs weakens and even reverses the correlation. To characterize these variables at different magnitudes, we calculated the Pearson correlation coefficient ($r$) and the squared coefficient of variance ($CV^2$) as measures of correlation and variability. We assumed that the $Rib^T$ variability is a constant (approximately 0.1, the variability lower bound of the typical abundant proteins in *E. coli* [25]). Here lies the explanation for the observed Simpson's paradox in multi-gene expression: the protein correlation is positive when $M_1^T$ variability is low (e.g., at each $P_1$ induction condition as a constant environment), which is dominated by the resource variation effect, but the correlation can be reversed by the competition effect at high $M_1^T$ variability (e.g., combining different $P_1$ induction conditions as a fluctuating environment) (Fig 2C and 2E). The contributions from the two variation sources to the protein correlation ($\frac{\partial P_1}{\partial Rib^T}\frac{\partial P_2}{\partial Rib^T}$ and $\frac{\partial P_1}{\partial M_1^T}\frac{\partial P_2}{\partial M_1^T}$) depend on the mean values of both $M_1^T$ and $Rib^T$ of the population (Note 1 in S1 Text). Intuitively, enhanced overexpression of heterologous genes (higher mean $M_1^T$) or limited total ribosome (lower $Rib^T$) would cause fewer resources to be devoted to expressing native genes in single cells, causing reduced correlation between competing proteins. In reality, our model shows that, within certain ranges (e.g., $M_1^T > 100$ and $Rib^T < 10,000$), a higher mean $M_1^T$ or a lower mean $Rib^T$ increases the relative contribution from $Rib^T$ variance compared with $M_1^T$ variance in Eq (1), leading to increased correlation between competing proteins (Fig 2C, 2D and 2F). These analyses are robust even when the full Eq (3) was used (S2C–S2J Fig).

Next, we investigated whether the Simpson's paradox also exists at the transcriptional level. We applied our model to transcriptional competition and solved for correlations between competing mRNAs in single cells (Note 2 in S1 Text and S3A Fig). The major difference between transcriptional and translational competition is that mRNA production was believed to be mainly determined by promoter strength (treated equivalently as promoter copy number in our model), and to a lesser extent, by the amount of RNAPs [26–28], so the effects of both RNAP competition and cell-to-cell variation in RNAPs are attenuated. Our model, with feasible parameters in transcription (i.e. the number total RNAP ranges from 4000 to 12000; dissociation constants for RNAP binding range from 0.1 to 10), predicts three phenomena: i)

within a large parameter range (1 to 100 copies of strong promoters per cell), introducing heterologous genes causes little repression on endogenous mRNA production (S3B Fig), ii) the correlations between competing mRNAs are determined by correlations between promoter strengths, and the promoter strength correlations can be weak or even negative in constant environments (S3C Fig), and iii) the correlations rarely change with promoter strength and its variability (S3D Fig). These features largely prevent the Simpson's paradox from occurring at the transcriptional level (mathematically explanation in Note 2 in S1 Text).

To validate model predictions, we experimentally quantified mRNA outputs of our testing modules in single cells, using two-color mRNA fluorescent in situ hybridization (FISH) (Fig 3A and 3B). The average GFP mRNA abundance was estimated to be approximately 2.02 ± 0.25 (mean ± s.d. across all conditions) copies per cell, ranking in the top 1% of all endogenous genes [25] and in agreement with RNA-seq measurements from the studied *E. coli* strain [29]. The GFP mRNAs at all induction levels followed similar Poisson distributions (S4 Fig), suggesting that endogenous mRNAs are not repressed by increasing heterologous mRNA levels (Fig 3C). Thus, both our model predictions and experimental results showed that resource competition mostly occurs at the translational level rather than at the transcriptional level, shining light on a previously debated issue about the cause of mRNA burden [7,29,30]. We further observed that the mRNA correlations in each induction condition were weak and positive, which also resulted in a weak and positive correlation when combining all conditions (Fig 3D). The result reveals that the strengths (or copy numbers) of these two promoter are weakly correlated likely due to cell division [31], and promoter strength variability with the RNAP competition effect alone is not sufficient to reverse the weak mRNA correlation in fluctuating environments.

Our data in Fig 1D showed that when expressing multiple genes under limited resources, the ratio of competing proteins in single cells varies even when they are growing in the same environments (e.g., induction levels). In some circumstances, such as expressing metabolic pathways or multi-protein complexes with precise stoichiometry, it is desirable to keep multiple genes expressed at a fixed ratio within single cells to achieve optimal overall performance and maximize the efficiency of resource utilization. Using polycistronic operons in combination with translational regulation is a common strategy for controlling the ratio of multiple proteins at the ensemble level [32,33]. However, the protein ratio in single cells may be affected by translational competition, resulting in disrupted stoichiometry. To examine the degree of competition effects on multi-gene expression from polycistronic operons in single cells, we constructed a library of polycistronic operons containing both *mCherry* and *gfp* genes driven by different promoters (Fig 4A). We found that the ratios of mCherry protein to GFP were consistent among single cells for each type of promoter, regardless of their promoter strength (Fig 4B and 4C). The ratios were observed to be different between the inducible $P_{LacUV5}$ promoter and constitutive promoters, which could be explained by different mRNA secondary structures near the ribosome binding site of the *mCherry* gene. In addition, the correlation between mCherry and GFP in single cells remained high, regardless of their expression strength and variability (Fig 4D and 4E). Collectively, these results suggest that resource competition and cellular heterogeneity hardly affect proportional protein production from the polycistronic operon.

Finally, we sought to explore the evolutionary benefits of correlated protein outputs in single cells in the presence of resource competition. We considered a generic horizontal gene transfer process, where the acquired genes bring beneficial functions, while they also negatively affect the expression of native genes by competing for limited resources. An antibiotic resistance model was built, where a species can independently deactivate two antibiotics by producing two resistance proteins, respectively (Note 3 in S1 Text). Positively correlated resistance proteins allow a small subpopulation of cells to survive high concentrations of both antibiotics

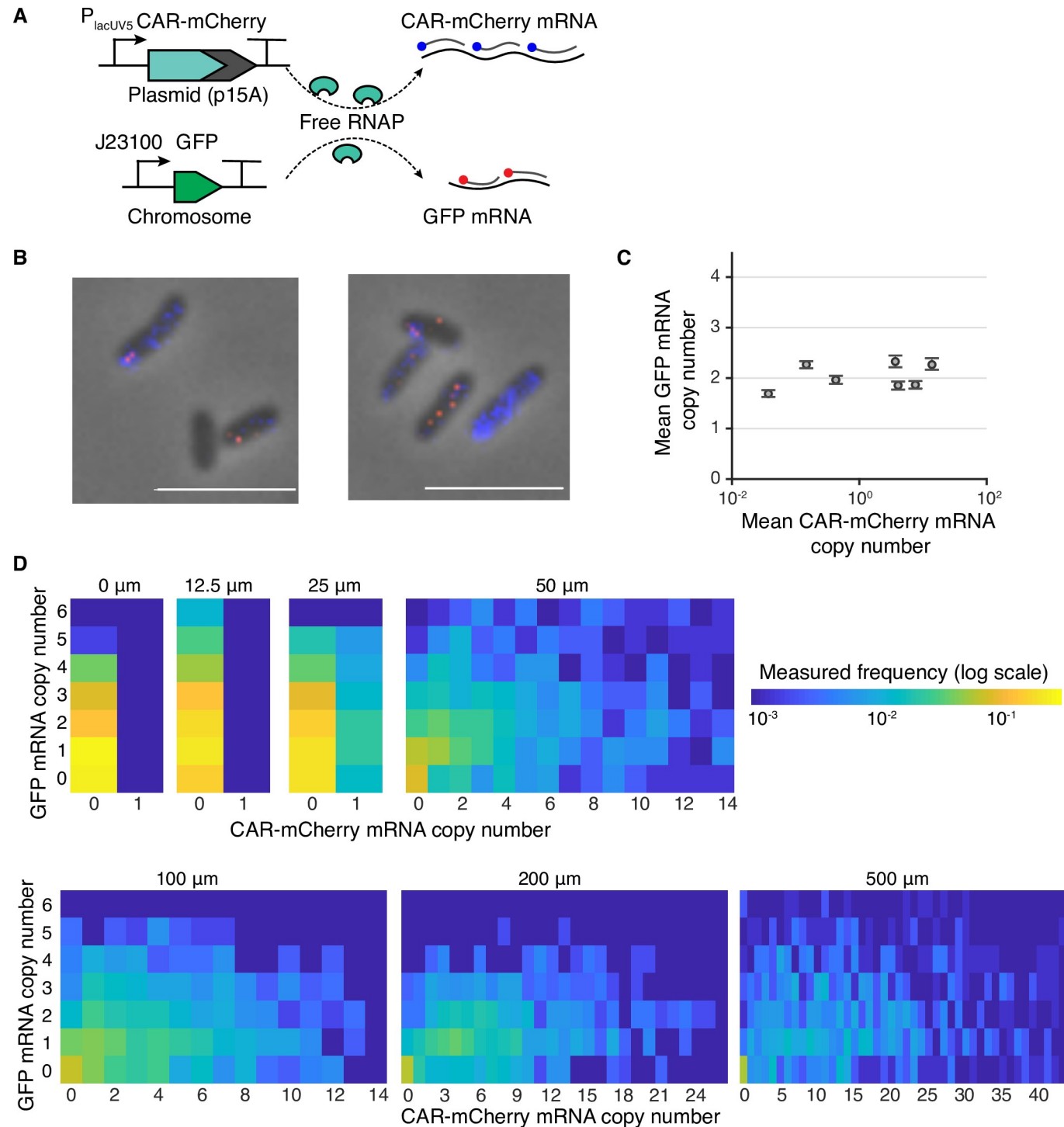

**Fig 3. Multi-gene expression in single-cells during transcriptional competition. (A)** Transcriptional competition between *car-mCherry* and *gfp* genes for limited shared RNAPs in single cells. CAR-mCherry mRNA and GFP mRNA were hybridized by Quasar 670- (blue) and Quasar 570-labeled (red) probes, respectively. The fluorescence of the mCherry protein was deactivated via the M71G mutation to prevent spectral overlap. **(B)** Representative FISH images of single cells induced at 500 μM IPTG. **(C)** Population mean mRNA copy numbers of GFP and CAR-mCherry at various IPTG concentrations. mRNA copy numbers of CAR-mCherry and GFP were estimated from fluorescence intensity. Error bars represent the 95% confidence interval, determined by bootstrapping. **(D)** GFP and CAR-mCherry mRNA copy numbers of single cells at various IPTG induction levels.

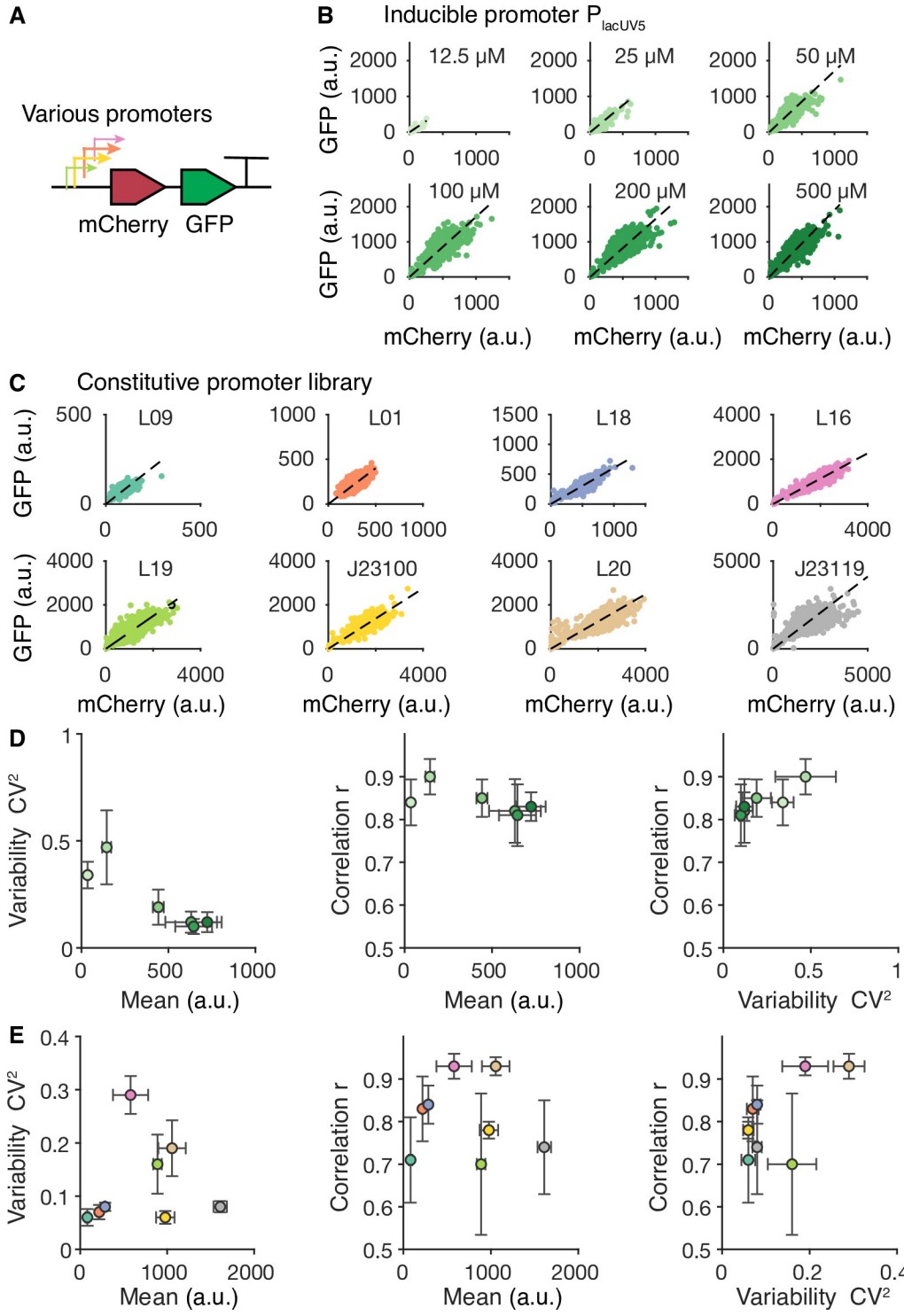

**Fig 4. Polycistronic operon enables highly correlated protein expression. (A)** Various promoters are used to control the co-expression of mCherry and GFP from a polycistronic operon. **(B)** mCherry and GFP in individual cells under the control of the inducible promoter $P_{lacUV5}$ at different IPTG induction levels. a.u., arbitrary units. **(C)** mCherry and GFP in individual cells under the control of constitutive promoters with different strengths. **(D)** Relationships among variability, mean, and correlation between mCherry and GFP in the inducible promoter construct. **(E)** Relationships among variability, mean, and correlation between mCherry and GFP in promoter library constructs. Variability and mean are quantified using GFP.

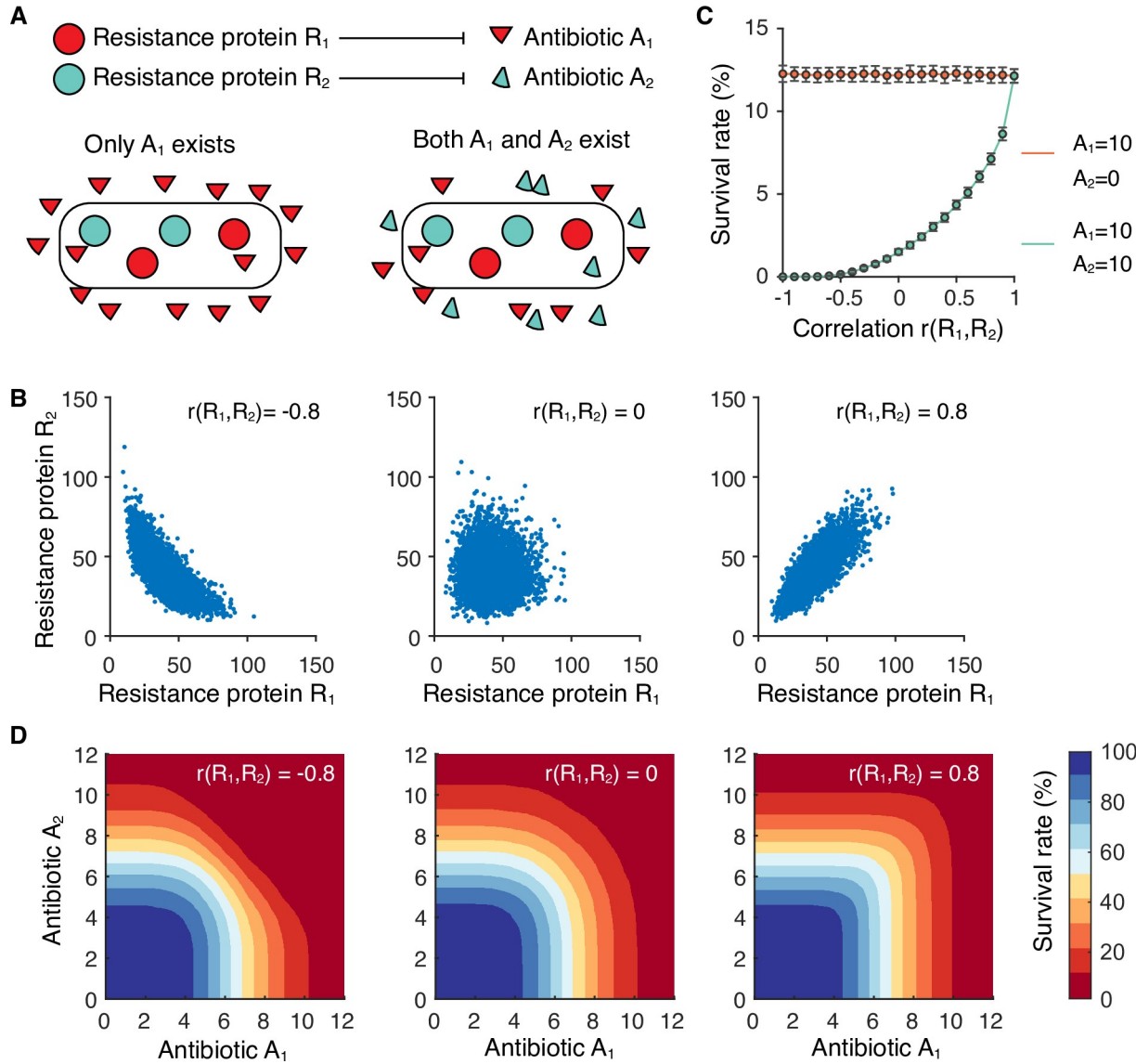

**Fig 5. Correlated expression of resistance proteins in single cells facilitates population survival under multiple antibiotics. (A)** An antibiotic resistance model. Two hypothetical antibiotics, $A_1$ and $A_2$, are independently deactivated by two resistance proteins, $R_1$ and $R_2$, respectively. Population survival rates are simulated in the presence of both $A_1$ and $A_2$. **(B)** Simulated joint distribution of $R_1$ and $R_2$ at three different scenarios: negative correlation with $r(R_1,R_2) = -0.8$, uncorrelated with $r(R_1,R_2) = 0$, and positive correlation with $r(R_1,R_2) = 0.8$. **(C)** The dependence of population survival rate on the correlation between $R_1$ and $R_2$. Error bars represent standard deviations of 100 simulations. **(D)** Survival rate profiles at three simulated correlations as in **B**.

(Fig 5), presenting a strategy for a population to cope with extremely harsh environments. Because the resource competition effect is always accompanied by resource variation, our results suggest an evolutionary mechanism that bacteria can use to compensate for the negative resource competition effect during horizontal gene transfer.

## Discussion

Overall, our results reveal that heterogeneity in shared resources and in competing modules are two seemingly opposite driving forces that work together to coordinate protein outputs for

all genes in single cells. In harsh environments, positively correlated protein outputs allow a small subpopulation of cells with abundant resources to support multitasking, facilitating individual survival and evolution of the population, which could present a previously unknown challenge in treating multi-drug resistant bacteria [34]. As a resource becomes abundant for all cells, the corresponding module outputs no longer depend on the amount of the resource. In this case, the effects of both resource competition and resource variation are weak, and the module outputs rely solely on the corresponding module inputs and thus function independently. This understanding of generic resource allocation in single cells provides a basis for analyzing and designing more sophisticated gene regulatory networks with high precision and ensemble efficiency.

Theoretically, our analytic framework can also be extended to describe competition and heterogeneity in other competing cellular processes. For example, two enzyme pathways often compete for a shared metabolite substrate. In this case, competition between two metabolic pathways, together with heterogeneity in cellular metabolite concentration, could affect single-cell metabolic flux in a similar way to that analyzed in this work, illuminating metabolic behavior previously unknown from existing analyses that do not consider their joint effects [14,35–37]. This improved understanding would bring us closer to more precise design of engineered microbial systems for various applications in biotechnology.

## Materials and methods

### Strains and DNA construction

The DH10GFP *E. coli* strain originally created by the Ellis lab [29] was ordered from Addgene (# 109392). The carboxylic acid reductase (*car*) gene was PCR amplified from the pB5k-sfp-car plasmid as described in previous work [16]. A mCherry gene was fused to the C-terminus of the *car* gene via a linker that encodes a helix-forming peptide A(EAAAK)$_3$A, as used in previous paper [29]. The *car*-mCherry fusion gene was cloned into a BglBrick vector pBbA5c (p15A origin, lacUV5 promoter, chloramphenicol selection marker) via Golden Gate DNA Assembly, resulting in plasmid pBbA5c-CAR-mCherry. Meanwhile plasmid pBbA5c-CAR-mCherry (M71G) carrying a non-fluorescent mCherry mutant (M71G) was created via site-directed mutagenesis and was used in FISH experiments. Plasmids pBbA5c-CAR-mCherry and pBbA5c-CAR-mCherry(M71G) were individually transformed to strain DH10GFP, yielding strains sYH006 and sYH013, respectively (S2 Table). *E. coli* DH10B strain was purchased from New England Biolabs Ltd. (Ipswich, MA, USA) and used as a negative control in the FISH experiment.

To investigate correlated protein expression from the same operon, an IPTG-inducible P$_{lacUV5}$ promoter and a library of constitutive promoters were used to control the transcription of mCherry and GFP from the same mRNA. Strong and identical RBS sequences (tttaagaaggagatatacat) were used for both mCherry and GFP. A small library of constitutive promoters (S1 Table) was designed based on the sequence of BioBrick promoter J23119, and was constructed into a plasmid with SC101 origin and chloramphenicol selection marker using a one-step Golden-Gate DNA Assembly. All plasmids were confirmed by Sanger sequencing.

### Growth conditions

Cell cultures were grown overnight in 3 mL of LB medium with 20 μg/mL chloramphenicol at 37˚C. The overnight cultures were diluted, in ratios between 1:400 and 1:1000, into 30 mL (for FISH samples) or 3 mL (for fluorescent protein assay samples) of M9 minimal medium, supplemented with 0.4% glucose, 1 mM thiamine, 0.4 mM leucine, and varying amounts of IPTG in either baffled shake flasks (for FISH samples) or test tubes (for fluorescent protein assay

samples). Cells were cultivated for approximately 10 hours (~5 cell cycles) and harvested in exponential phase when an $OD_{600}$ of 0.2–0.4 was reached. Cells cultivated for 9 hours to 12 hours were randomly harvested as controls to confirm that 10 hours incubation is enough for the cells to reach a steady state.

## Maturation of fluorescent proteins

To allow maturation of fluorescent protein for more accurate quantification, cells were incubated for an additional period before taking fluorescence measurements [38–40]. Specifically, 1 mL of cell cultures were transferred into pre-chilled test tubes and placed in ice-water bath for 10 min to halt cell growth and gene expression. The cell cultures were centrifuged at 13,000 rpm for 30 s at 4˚C. The supernatant was removed, and the pellet was resuspended in 1 mL of phosphate buffered saline (PBS) solution containing 500 µg/mL of rifampicin. The resuspended cells were incubated at 37˚C for 90 min and subjected to imaging.

## mRNA fluorescence *in situ* hybridization (FISH)

**Probe design.**   Two sets of custom probes for GFP and CAR-mCherry were designed using the online Stellaris Probe Designer (S4 Table) and synthesized by Biosearch Technologies Inc (Novato, CA, USA). Probes for GFP and CAR-mCherry were labelled with Quasar 570 and Quasar 670 fluorescent dyes, respectively.

**Fixation and labelling.**   Cell fixation and mRNA labelling were performed following established protocols[41]. In detail, 15 mL of each cell culture at $OD_{600}$ = 0.4 were collected and transferred to an ice-chilled 50-mL centrifuge tube, followed by immediate centrifugation at 4,500 g for 5 min at 4˚C. The supernatant was carefully removed, and the pellet was resuspended in 1 mL of 3.7% formaldehyde in 1x PBS. The resuspended cells were then mixed gently at room temperature for 30 min using a nutator. Next, the cells were centrifuged at 400 g for 8 min at room temperature, then washed twice with 1 mL of 1x PBS. Then the cells were resuspended in 300 µL of DEPC-treated water, permeabilized by adding 700 µL of 100% ethanol, and mixed for 1 hour at room temperature using a nutator. After mixing, the cells were centrifuged at 600 g for 7 min at room temperature, and then resuspended in 1 mL of 40% wash solution (353 µL formamide, 100 µL 20x saline-sodium citrate (SSC), 547 µL water). The resuspended solution was then gently mixed for 5 min at room temperature using a nutator and centrifuged at 600 g for 7 min at room temperature. For each sample, the cell pellets were resuspended in 50 µL of 40% hybridization solution (1 g of dextran sulfate, 3530 µL of formamide, 10 mg of *E. coli* tRNA, 1 mL of 20x SSC, 40 µL of 50 mg/mL BSA, and 100 µL of 200 mM ribonucleoside vanadyl complex for 10 mL solution) with probes at a final concentration of 1 µM per probe set. The mixture was incubated at 30˚C overnight. Samples after hybridization were then washed four times in 40% wash solution before imaging in 2x SSC.

## Microscopy and image analysis

Microscopy was performed using a Nikon Eclipse Ti microscope (Tokyo, Japan) equipped with an EMCCD camera (Photometrics Inc. Huntington Beach, CA, USA) and a 100 x, NA 1.40, oil-immersion phase-contrast objective lens. An X-Cite 120 LED was the light source. Three band-pass filter cubes (FITC, DsRed, and C-FL CY5, all from Nikon Inc.) were used for spectral separation. In both FISH and protein fluorescence experiments, an exposure time of 20 ms was used for phase-contrast images. In FISH experiments, the DsRed filter and the C-FL CY5 filter were used to detect Quasar 570 (exposure time of 500 ms, with an electro-multiplier gain of 200 x) and Quasar 670 (exposure time of 300 ms, with an electro-multiplier gain of 100 x), respectively. In protein fluorescence experiments, the FITC and the DsRed filter cubes were

used to detect GFP (exposure time of 500 ms, no electro-multiplication) and mCherry (exposure time of 300 ms, no electro-multiplication), respectively. The power of the LED light was carefully controlled so that no significant photobleaching was detected. Images were collected by an automated scanning function of the microscope with a built-in Perfect Focus System (PFS) and analyzed using the Nikon NIS-elements software package. On average, 3000 single cells per protein sample and 1000 single cells per FISH sample were collected and analyzed.

**Cell segmentation.** The phase-contrast images were used for cell identification and segmentation. Overlapped cells, dividing cells, and long unhealthy cells (totaling less than 1%) were excluded by a length filter, an area filter, and visual inspection.

**mRNA fluorescence quantification.** Single cell mRNA fluorescence was quantified following the previous method[41]. Specifically, background fluorescence was first subtracted to eliminate the effects of autofluorescence on different images. The total fluorescence intensity within a cell was normalized by the cell area to reduce the influence from variations in cell cycles and growth rates. False-positive thresholds for Quasar 570 and Quasar 670 were determined by the fluorescence distribution in a negative control sample (*E. coli* DH10B strain). The fluorescence intensity of a single mRNA was identified by the peak position of the fluorescence distribution in low-expression cells. To convert the total fluorescence in a cell to the mRNA copy number, we divided the total by the average fluorescence intensity of a single mRNA and rounded the value to the closest integer.

**Protein fluorescence quantification.** The background fluorescence of each image was subtracted, and the total fluorescence intensity of each cell was normalized by cell area. The cell-area-normalized total pixel intensity was used as the single-cell protein expression level.

## Statistics

Gene expression variability was quantified in terms of the variance over the squared mean. The Pearson correlation coefficient $r(X_1, X_2) = \frac{Cov(X_1, X_2)}{\sqrt{Var(X_1)Var(X_2)}}$ was utilized to quantify the correlation between the expression levels of two genes in single cells. The 95% confidence intervals of all estimated parameters were constructed by bootstrap method.

## Data and code availability

The data and the MATLAB codes for modelling results that support the findings are available from https://github.com/yhan0410/Data-and-model-in-Heterogeneity-coordinates-bacterial-multi-gene-expression-in-single-cells.

## Supporting information

**S1 Table. Sequences of constitutive promoters.**
(DOCX)

**S2 Table. Strains used in this study.**
(DOCX)

**S3 Table. Statistics determined by single cell experiments in this work.**
(DOCX)

**S4 Table. Probes used in FISH experiments.**
(DOCX)

**S1 Text. Models and parameters.**
(DOCX)

**S1 Fig. Data reproducibility for the Simpson's paradox phenomenon in multi-gene expression.** (A) Dashed lines are linear fitting of the merged data. The three replicates were performed at different days. (B) Correlation from random and evenly sampling across all induction conditions. Error bars represent standard deviations of 100 replicates.
(TIF)

**S2 Fig. Translational resource competition under various parameters.** (A) The relationship between endogenous protein ($P_2$) and heterologous proteins ($P_1$) at various $\beta_1$ values. (B) The relationship between endogenous protein ($P_2$) and heterologous proteins ($P_1$) at various $\beta_2$ values. $\beta_1$ and $\beta_2$ are varied by tuning $\beta_1^+$ and $\beta_2^+$ (from $1^*10^{-2}$ to $1^*10^{-6}$) respectively. (C-J) The same relationship as Fig 2C–2F with $M_2^T$ variability set as 0.1. (C-F) correlation between $M_1^T$ and $M_2^T$ is set as 0. (G-J) correlation between $M_1^T$ and $M_2^T$ is set as 0.2.
(TIF)

**S3 Fig. Coarse-grained model of transcriptional resource competition.** (A) Schematic of RNAP allocation among heterologous genes ($i = 1$), endogenous protein-coding genes ($i = 2$), and rRNA/tRNA genes ($i = 3$). $RNAP^F$, free RNAPs; $RNAP^T$, total RNAPs; $D_i^F$, genes free from RNAPs; $D_i^C$, gene-RNAP complexes; $D_i^T$, total genes; $M_i$, total mRNAs. (B) RNAP competition in a single cell. Left, relationship between $D_1^T$ and the fraction of free RNAP ($RNAP^F/RNAP^T$). Right, relationship between heterologous mRNA ($M_1$) and endogenous mRNA ($M_2$) caused. Calculations of $RNAP^F$, $M_1$, and $M_2$ are described in Note 2.2 in S1 Text with parameters listed in Table A in S1 Text. (C) Correlations between competing mRNAs in single cells r ($M_1$, $M_2$) changes with correlations between promoter strengths r($D_1^T$, $D_2^T$) (left), $D_1^T$ (center), and $RNAP^T$ (right). $D_1^T > 200$ is considered as unrealistic region. $RNAP^T$ affects r($M_1$, $M_2$) only in RNAP limiting region.
(TIF)

**S4 Fig. Distributions of mRNA copy number under different conditions.** Single-cell GFP mRNA copy numbers measured from FISH were fitted to Poisson distributions due to its transcription from a constitutive promoter. CAR-mCherry mRNA copy numbers were fitted with negative binomial distributions because they were transcribed from an inducible promoter.
(TIF)

## Acknowledgments

We thank A. Schmitz, C. Hartline, C. Sargent, and J. Ballard for discussion and technical assistance.

## Author Contributions

**Conceptualization:** Fuzhong Zhang.

**Data curation:** Fuzhong Zhang.

**Formal analysis:** Yichao Han, Fuzhong Zhang.

**Funding acquisition:** Fuzhong Zhang.

**Investigation:** Yichao Han, Fuzhong Zhang.

**Methodology:** Yichao Han, Fuzhong Zhang.

**Project administration:** Fuzhong Zhang.

**Resources:** Fuzhong Zhang.

**Software:** Yichao Han.

**Supervision:** Fuzhong Zhang.

**Writing – original draft:** Yichao Han, Fuzhong Zhang.

**Writing – review & editing:** Yichao Han, Fuzhong Zhang.

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
