## [Decision Letter · Decision Letter 0]

1 Nov 2019

Dear Dr Zhang,

Thank you very much for submitting your manuscript 'Heterogeneity coordinates bacterial multi-gene expression in single cells' for review by PLOS Computational Biology. Your manuscript has been fully evaluated by the PLOS Computational Biology editorial team and in this case also by independent peer reviewers. The reviewers appreciated the attention to an important problem, but raised some substantial concerns about the manuscript as it currently stands. While your manuscript cannot be accepted in its present form, we are willing to consider a revised version in which the issues raised by the reviewers have been adequately addressed. We cannot, of course, promise publication at that time.

Sincerely,

Christoph Kaleta

Associate Editor

PLOS Computational Biology

Alice McHardy

Deputy Editor

PLOS Computational Biology

[LINK]

Reviewer's Responses to Questions

**Comments to the Authors:**

Reviewer #1: The article presents experimental and modeling work on the correlation between protein levels (and between RNA levels) in a population of genetically identical bacteria. The central finding concerns the relationship between the anti-correlation of protein levels that occurs when different conditions are merged and the positive correlation that characterizes each condition individually. Fluctuations in available resources (mRNAs in particular) are identified as the key to explain observations. On the plus side I found the paper quite well written. The modelling results and experimental data are presented clearly and thoroughly. On the minus side, the modeling framework needs to be placed in the context of existing models, and the interpretation of the main finding is in my view not completely convincing.

Main:

1) The general problem faced in this work (competition, allocation of cellular resources etc.) is well studied and the model discussed in this paper seems to be close to previous models both technically and in terms of the questions being asked and lessons being drawn (the fact that P1 is considered as “heterologous” seems to me to be immaterial for the conclusions). For instance, the fact that (the expression levels of) two proteins display negative correlation (in any given condition) immediately suggests, as the authors say, competition for shared positive regulators (eg ribosomes). The full picture is however much richer and includes the possibility of having positive correlations, depending essentially on kinetic details (see e.g. 10.1016/j.bpj.2013.04.049 & doi.org/10.1371/journal.pcbi.1002203). On the other hand a positive correlation can also be induced by competition for a shared negative regulator of gene expression such as microRNAs (acting on mRNAs, see e.g. 10.1038/srep43673 ). (The suggested links only represent a few examples that came to my mind, but the modelling literature on these topics is huge.) In each case, correlations between the corresponding transcripts do not need to reflect those between their functional products. In my view, a discussion of previous approaches and of how the present model deviates from/generalizes/complements them is necessary. In particular, it would be important that authors clarify what biological insight discussed in this paper cannot be obtained without the specific modeling frame/assumptions they employed. In this respect, I think that the role of resource variability could be further highlighted against previous work.

2) Regarding the Simpson paradox, merging data coming from different conditions does not necessarily yield, as far as I understand, a new condition from which conclusions can be drawn. In general I would say that cases like the experiment of Hecht & al [S Hecht, S Shlaer, and MH Pirenne, Energy, quanta and vision. , J Gen Physiol 25, 819–840 (1942)] provide a strong caveat against doing it: averaging over different conditions (patients in their case) can lead to erroneous interpretations of data. I understand that the authors take the merged dataset to model a “heterogeneous and fluctuating” environment, but frankly I am not convinced. Looking at individual conditions one would conclude that the competition between the two proteins is not there and everything is driven by the induction that changes the slope of the protein-protein dependence across different conditions. The fact that mixing experiments one observes a negative correlation does not change this fact. So why exactly do authors deem it important/interesting that averaging over conditions the correlation changes sign? Why is this special? This is really not clear to me, also because the negative correlation of mean values seems to be rather weak (especially when compared against the range of variability of single cells).

Important: it seems to me (Table S3) that the nr of cells is rather unevenly distributed across induction conditions (less cells at the maximum level compared to the control). Am I right in assuming that the fitting procedure used for the merged data accounts for this imbalance? (Otherwise the fit could be biased to return a negative correlation). This should be made very clear all throughout the text.

Minor:

Supplementary Note: The first equation (unnumbered) of Note 1 as well as the first equation (unnumbered) of Note 2 appear on a single line without any separation in my doc reader. This is confusing (but it may depend on the reader, not sure)

About parameters: Models tend to use somewhat standardised parameters so I don’t doubt that the representative results displayed by the authors represent a physiologically realistic scenario. However a discussion of how sensitive results are to the model’s parameters would be welcome.

Equation 1 plays a central role in this manuscript. The approximation based on which it is derived (Supplementary Note 1) seems reasonable to me but I would stress it in the Main Text. Also, Eq 1 is rather intuitive once the approximation is explained. I suggest the authors provide the reader with some guideline to interpret the physical meaning of Eq 1 already in the Main Text.

Reviewer #2: The work by Han and Zhang reports an extremely interesting study on resource competition in single cells at the translational and transcriptional level. The authors found that the former correlates positively in single cells, but negatively at in the population. This Simpson paradox is not found at the transcriptional level.

The work is very nicely and concisely summarized. I only have some minor suggestions

* as the authors submitted to PLOS CB, I do not think the mathematical model needs to be hidden in the supplementary materials. Rather the model and major mathematical results should be explicitly shown in the main text. I believe this is particularly true for line 80 to 130, where the train of thought is interrupted by reference to the supplementary material.

* the authors say within certain ranges (line 107) with feasible parameters (line 116-117). I think these number should be made explicit (including some discussion) in the main text.

* the connection between mathematical model and experimental realization may be improved if Fig 1A for instance also includes the model variables.

* the authors say shining light on a previously debated issue (line 131-132). Please, could you briefly indicate the arguments put forward by the references in this debate.

* line 134 - 136, what would be required to reverse the correlation so that’s consistent with line 119?

* please deposit your data and matlab scripts on github or some other public database

**Have all data underlying the figures and results presented in the manuscript been provided?**

Reviewer #1: Yes

Reviewer #2: No: Some data, in particular the scripts are only available on requests

PLOS authors have the option to publish the peer review history of their article (what does this mean?). If published, this will include your full peer review and any attached files.

Reviewer #1: No

Reviewer #2: No

---

## [Decision Letter · Decision Letter 1]

9 Jan 2020

Dear Dr Zhang,

We are pleased to inform you that your manuscript 'Heterogeneity coordinates bacterial multi-gene expression in single cells' has been provisionally accepted for publication in PLOS Computational Biology.

In the meantime, please log into Editorial Manager at https://www.editorialmanager.com/pcompbiol/, click the "Update My Information" link at the top of the page, and update your user information to ensure an efficient production and billing process.

One of the goals of PLOS is to make science accessible to educators and the public. PLOS staff issue occasional press releases and make early versions of PLOS Computational Biology articles available to science writers and journalists. PLOS staff also collaborate with Communication and Public Information Offices and would be happy to work with the relevant people at your institution or funding agency. If your institution or funding agency is interested in promoting your findings, please ask them to coordinate their releases with PLOS (contact ploscompbiol@plos.org).

Thank you again for supporting Open Access publishing. We look forward to publishing your paper in PLOS Computational Biology.

Sincerely,

Christoph Kaleta

Associate Editor

PLOS Computational Biology

Alice McHardy

Deputy Editor

PLOS Computational Biology

Reviewer's Responses to Questions

**Comments to the Authors:**

Reviewer #1: My concerns have been addressed. The connection between the merged dataset and heterogeneous regions in extended systems makes is indeed helpful.

Reviewer #2: my concerns have been appropriately addressed

**Have all data underlying the figures and results presented in the manuscript been provided?**

Reviewer #1: None

Reviewer #2: Yes

PLOS authors have the option to publish the peer review history of their article (what does this mean?). If published, this will include your full peer review and any attached files.

Reviewer #1: No

Reviewer #2: No

---

## [Editor Report · Acceptance letter]

23 Jan 2020

PCOMPBIOL-D-19-01269R1 

Heterogeneity coordinates bacterial multi-gene expression in single cells

Dear Dr Zhang,

I am pleased to inform you that your manuscript has been formally accepted for publication in PLOS Computational Biology. Your manuscript is now with our production department and you will be notified of the publication date in due course.

With kind regards,

Sarah Hammond
